

# Analytical and numerical study of the salinity intrusion in the Sebou river estuary (Morocco). Effect of the 'Super Blood Moon' (total lunar eclipse) of 2015

Soufiane Haddout*, Mohammed Igouzal, Abdellatif Maslouhi

Interdisciplinary Laboratory for Natural Resources and Environment, Department of Physics,  Faculty of Sciences, Ibn Tofail University, B.P 242, 14000 Kenitra, Morocco
*Correspondence to: Haddout. S (haddout.ens@gmail.com)

**Abstract.** The longitudinal variation of salinity and the maximum salinity intrusion length in an alluvial estuary are

important environmental concerns for policy makers and managers since they influence water quality, water utilization and agricultural development in estuarine environments and the potential use of water resources in general. Total eclipses of Super-Moons are rare. According to NASA, they have only occurred five times in the 1900s-in 1910, 1928, 1946, 1964 and 1982. After the September 28th, 2015 Total lunar eclipse, a Super-Blood-moon eclipse will not recur before October 8th, 2033. In this paper, for the first time, the impact of the total lunar eclipse (Super Blood Moon) on the salinity intrusion along

an estuary is studied. The 28th September 2015 total lunar eclipse is focused by the study and the Sebou river estuary (Morocco) is taking as an application area. The Sebou estuary is an area with high agricultural potential, is becoming one of the most important industrial zones in Morocco and it is experiencing a salt intrusion problem. Hydrodynamic equations for tidal wave propagation coupled with (Savenije theory), and a numerical salinity transport model (HEC-RAS) are applied to study the impact of the total lunar eclipse on the salinity intrusion. Intensive salinity measurements during this extreme event

were recorded along the Sebou estuary. Measurements showed a modification of the shape of axial salinity profiles and a notable water elevation rise, compared with normal situations. The two optimization parameters (Van Der Burgh's and dispersion coefficients) of the analytical model are estimated based on the Levenberg-Marquardt´s algorithm (i.e. solving non-linear least-squares problems). The salinity transport model was calibrated and validated using field data. The results show that the two models described very well salt intrusion during the total lunar eclipse day. A good-fit between computed

salinity and measurements is obtained, as verified by statistical performance tests. These two models can give a rapid assessment of salinity distribution and consequently help to ensure the safety of water supply, even during such infrequent astronomical phenomenon.

**Key -words:** 2015's total lunar eclipse; salinity intrusion; hybrid model; Savenije theory; HEC-RAS;  Sebou  estuary.





## 1. Introduction

A total lunar eclipse is one of Nature's loveliest celestial events (Espenak, 2000). Any total lunar eclipse offers unique and breathtaking wonders; the full moon becomes not just a red "blood moon" but also is a Harvest Moon and a "super moon".

Total lunar eclipses appear red and for this reason are often called blood moons (Fig.1).

Tidal motions arise as a response to forces associated with the interaction of the earth-moon-sun system, the effect of the moon being about twice that of the sun (Stronach, 1989). A total lunar eclipse occurs when the moon enters the cone of shadow cast by the sunlit earth (i.e. the sun and the moon are in line with the earth) (Fig.2). This shadow covers the entire moon and causes a total lunar eclipse. The high tide is at its highest point and the low tide at its lowest point, the

gravitational pull on the ocean is then strong. Lunar eclipses are relatively infrequent events and on average only two of them may be contemplated each year (Mallama, 1996 , Muñoz and Pallé, 2011).

Estuaries form essential parts of the human-earth system (Savenije, 2015). As the connecting element between marine water and river, estuaries have properties of both: they contain both fresh and saline water; they experience tides, but also river floods; and they host both saline and fresh ecosystems (Savenije, 2015). The diverse estuarine environment plays an

important role in the life cycle of many species but also serves as a site for many human activities. In short, estuaries are important water bodies where many dynamic factors interact and unfold (Xu et al., 2015). For decades, explosive increases in industrial and agricultural productivity, as well as the growing population in estuary regions, have led to numerous environmental concerns (Mai et al., 2002). Salinity intrusion is an important phenomenon in an estuary, and can constitute a serious problem. It influences the water quality and threatens potential water resource use. Intake of fresh water for

consumption, agricultural purposes or use by industries may take place within a region not far landward of the limit of salt intrusion. To support policy and managerial decisions, a profound knowledge of processes associated with the salinity structure in estuaries is required (Kuijper and Van Rijn, 2011). Models have been widely used to research salinity intrusion. Two kinds of models are typically used: numerical models and analytical models. Presently, numerical models are more popular especially 2-D and 3-D models (Kärnä et al., 2015; Elias et al., 2012; Zhao et al., 2012; Li et al., 2012; Jeong et al.,

2010; Wu and Zhu, 2010; Xue et al., 2009; An et al., 2009, etc.), because they can provide more spatial and temporal detail. Analytical models are also widely used, such as Prandle (1981), Savenije (1986, 1989, 1993a, 2005, 2012, 2015), Lewis and Uncles, 2003, Gay and O' Donnell (2007, 2009), Kuijper and Van Rijn (2011), Aertsl et al., (2000), Brockway et al., (2006), Nguyen and Savenije (2006), Nguyen et al., (2008a, 2008b), Cai et al., (2015a), Gisen et al., (2015a), Gisen et al., (2015b), Xu et al., (2015). These tools are based on the steady-state conservation of mass equation which indicates that the dispersive

and advective transports of salt are in equilibrium and the effective longitudinal dispersion coefficient incorporates all mixing mechanisms, where the dispersion coefficient along the estuary axis is either constant (e.g., Gay and O'Donnell, 2007) or variable (e.g., Van der Burgh, 1972; Savenije, 1986).



One-dimensional mathematical models (analytical or numerical) can constitute the appropriate tools for quick-scan actions in a pre-phase of a project or for instructive purposes. Also, it is methodologically correct to start with the simplest description

of the phenomena under study and to evaluate the limits of this approximation before investigating more complications.

Our previous studies on the Sebou estuary have shown that the one-dimensional (analytical or numerical) methods compute properly salt intrusion (Haddout et al., 2015a, Haddout et al., 2015b). In addition, Haddout et al., 2015b showed that salinity profiles of Sebou estuary show steep decrease. This characteristic is specific to narrow (recession shape) estuaries (i.e. having a near-prismatic shape and significant freshwater discharge). Such estuaries are called positive estuaries.

The aims of this paper it to investigate the applicability of these two methods (analytical or numerical) during the Super-moon day of 28$^{th}$ September 2015 (total lunar eclipse). Measurements showed a modification of the form of the salinity profiles along the estuary and a notable water level increase, compared with normal situations studied in our earlier works. In addition, calculations during lunar eclipse using the coupled analytical hydrodynamic-salt intrusion model required the recalculation of the geometric parameters of the estuary i.e., cross-sectional areas $A_0$, convergence lengths a.

Even in these extreme conditions, a good fit was obtained between the computed and observed salinity distribution for the two models. These models constitutes powerful tools for evaluating salinity intrusion pattern in the Sebou river estuary, even during extreme events inducing sea level rise like lunar eclipse or climate change.

## 2. Models formulations

**1.2. One-dimensional salt intrusion model**

The analytical salinity intrusion model of Savenije (2005) has been adopted to predict the salinity distribution and salinity intrusion length in alluvial estuaries. This method is fully analytical, although it makes use of certain assumptions, the most important being: the exponential shape of the estuary, the longitudinal variation of the dispersion according to Van der Burgh (1972), and the predictive equations for the boundary condition and the Van der Burgh coefficient. The equations are

based on the 1-D cross-sectionally averaged, and tidally averaged, steady-state salt balance equation, in which the advective salt transport is caused by the downstream fresh water discharge, counteracted by the landward dispersive salt transport induced by the different mixing processes (Cai et al., 2015b). In a convergent estuary, where the main geometric parameters (cross-sectional area $A$, width $B$ and depth $h$) can be described by exponential functions (See equations (2.38)-(2.40) in Savenije, 2012).

In a steady-state situation, the partial temporal derivative in the salt balance equation is zero (Gisen et al., 2015b). Considering constant fresh water discharge $Q_f$ and tidally averaged cross-sectional area A, the salt balance equations for High Water Slack (HWS), Low Water Slack (LWS) and Tidal Average (TA) situation can be rearranged as:





$$\frac{\partial S}{S} = -\frac{|Q|}{A.D}\partial x \tag{1}$$

If one assumes that D is constant along the length of an estuary (i.e., D=$D_0$, where $D_0$ is the dispersion coefficient at the
estuary mouth), simple analytical solutions for axial distribution of salinity can be derived.

Since dispersion depends both on the river discharge and the salinity distribution itself, the constant dispersion is not a
correct solution. An efficient and accurate approach to simulate the longitudinal variation of dispersion is presented by Van
der Burgh (1972), and also adopted by Savenije (1986, 1989, 1993a, 2005, 2012):

$$\frac{\partial D}{\partial x} = -K\frac{|Q|}{A} \tag{2}$$

Which, using equation (1), can be demonstrated to be the same as the following equation (Savenije, 2005, 2012):

$$\frac{D}{D_0} = \left[\frac{S}{S_0}\right]^K \tag{3}$$

where $S_0$ (g/l) is the boundary salinity at the estuary mouth, $D_0$ (m²/s) is the longitudinal dispersion at the estuary mouth for
HWS, LWS or TA condition, $K$ is the Van der Burgh's dimensionless coefficient, which has a value between 0 and 1. If $K$
= 0, equation (3) reduces to the case with constant dispersion D=$D_0$. For the case where $K$ =1, we see that the curves $D/D_0$
and $S/S_0$ coincide. If $K$ is small, then tide-driven mixing is dominant near the toe of the intrusion curve; if $K$ approaches
unity, then gravitational mixing is dominant (Savenije, 2006; Shaha and Cho, 2011).

Integrating equation (3) in combination with (equation (2.38) see in Savenije (2012)) yields:

$$\frac{D}{D_0} = \left[1 - \frac{K.|Q|.a}{A.D_0}\left\{\exp(x/a) - 1\right\}\right] \tag{4}$$

Which determines the longitudinal variation of dispersion coefficient. Combining equations (3) and (4), the cross-sectionally
averaged salinity along an estuary with convergent cross-sectional area at HWS is given by (Savenije, 2005, 2012):

$$\frac{S^{HWS}}{S_0} = \left[1 - \frac{K.a.|Q|}{D_0^{HWS}.A_0}\left(\exp(x/a) - 1\right)\right]^{1/K} \tag{5}$$



Making use of the dimensionless parameters (Cai et al., 2015a), equation (5) can be scaled as:

$$S^* = \left[ 1 - \frac{K.D^*}{\gamma} \left( \exp(x^* \gamma) - 1 \right) \right]^{1/K} \tag{6}$$

Where: $S^*$ is dimensionless salinity that is normalized by the salinity at the estuary mouth, $\gamma$ is the estuary shape number representing the convergence of an estuary, $D^*$ is the dimensionless dispersion at downstream boundary condition and $x^*$ is the dimensionless longitudinal coordinate that is scaled by the frictionless wave length in prismatic channels (Cai et al., 2015a).

10 This one-dimensional steady advection-diffusion model has been applied to describe the salinity distribution along numerous well-mixed and partially-mixed estuaries (i.e. the adopted salt intrusion model assumes a partially to well mixed situation, which under low flow is the dominant process) (Nguyen and Savenije, 2006) for HWS, LWS or TA condition.

The salt intrusion length $L^*$, defined as the distance from the estuary mouth to the location with fresh water salinity (assumed to be <1g/l isohaline) for HWS, LWS or TA condition, can be determined by setting $S^* = 0$ in equation (6):

$$L^* = \frac{1}{\gamma} \ln(\frac{\gamma}{D^* K} + 1) \tag{7}$$

On other hand, in most estuaries, there usually exists an inflection point near the mouth, where the geometry changes (e.g., Gisen et al., 2015b). This inflection point is associated with the transition of a wave-dominated regime to a tide-dominated regime (Gisen et al., 2015b). Making use of this phenomenon, Gisen et al. (2015b) recently expanded the underlying data
20 base and reanalyzed these equations, resulting in:

$$D_{1(\text{Revised})} = v_1 E_1 \left[ 0.396.(g / C^2)^{0.21} (\text{N}_R)^{0.57} \right] \tag{8}$$

$$K_{\text{Revised}} = 8.03 \times 10^{-6} \left[ \frac{B_f^{0.30} g^{0.93} H_1^{0.13} T^{0.97} \pi^{0.71}}{B_1^{0.30} C^{0.18} v_1^{0.71} b_1^{0.11} h_1^{0.15} r_s^{0.84}} \right] \tag{9}$$





Where: $B_f$ is the river regime width (typical width in the river dominated region), and $C$ is the Chezy roughness coefficient.

The symbols $H_1$ (m); $B_1$ (m); $v_1$ (m/s); $b_1$ (m); $h_1$ (m); $D_1$ (m$^2$/s); $E_1$ (m) represent the tidal range, stream width, velocity amplitude, width convergence length, depth, dispersion coefficient, and tidal excursion at the inflection point, respectively. If there is no inflection point near the estuary mouth, then these parameters refer to the situation at the mouth itself.

The roughness coefficient can be estimated by $C = K_s h_1^{1/6}$ with $K_s$ being the Manning-Strickler friction coefficient, while the tidal excursion can be calculated by $E_1 = v_1 T / \pi$. The Estuarine Richardson number $N_R$ (-) which is defined as the ratio of potential energy of the buoyant fresh water to the kinetic energy of the tide (Fischer et al., 1979) is given by:

$$N_R = \frac{\Delta\rho}{\rho} \cdot \frac{g.h.|Q|.T}{A_0.E_0.v_0^{\,2}} \tag{10}$$

Where $\rho$ (kg/m$^3$) is the water density, $\Delta\rho$ is the density difference of ocean and river water over the salt intrusion length (in estuaries, the ratio $\Delta\rho / \rho$ is about 0.025).

## 1.1. Analytical hybrid (hydrodynamic) model

Since 1960-s there exists a long tradition of one-dimensional analytical solutions for tidal dynamics in estuaries (e.g., Dronkers, 1964; Ippen, 1966; Prandle and Rahman, 1980; Leblond, 1978; Godin, 1985, 1999; Jay, 1991; Friedrichs and Aubrey, 1994; Lanzoni and Seminara, 1998; Kukulka and Jay, 2003; Horrevoets et al., 2004; Jay et al., 2011; Cai et al., 2012a). These analytical solutions usually made assumptions to simplify or linearize the non-linear set of equations (Zhang et al, 2012). Of these, most authors used perturbation analysis, where scaled equations are simplified by neglecting higher order terms, generally neglecting the advective acceleration term and linearizing the friction term, higher-order terms, whereas Savenije (2005) uses a simple harmonic solution without simplifying the equations (Cai et al., 2013). Others used a regression model to determine the relationship between river discharge and tide. Exceptions are the approaches by Horrevoets et al. (2004) and Cai et al. (2012b), who provided analytical solutions accounting for river discharge, based on the envelope method originally developed by Savenije (1998) (Cai et., 2013). Recently, Cai et al. (2012a) proposed a new analytical framework for understanding the tidal damping in estuaries. They concluded that the main differences between the examined models (e.g., Savenije et al., 2008; Toffolon and Savenije, 2011; Van Rijn, 2011) lies in the treatment of the friction term in the momentum equation. Furthermore, Cai et al. (2012a) presented a new 'hybrid' expression for tidal damping as a weighted average of the linearized and fully non-linear friction term (Cai et al., 2013). Additionally, Cai et al. (2014b) included for the first time the effect of river discharge in a hybrid model that performs better.





It can be demonstrated that tidal hydrodynamics is controlled by three dimensionless parameters that depend on localized geometry and external forcing (e.g., Toffolon et al., 2006; Savenije et al., 2008), i.e.: ζ the dimensionless tidal amplitude (indicating the seaward boundary condition), γ the estuary shape number (representing the effect of cross-sectional area convergence) and χ the friction number (describing the role of the frictional dissipation). These parameters are defined in Table 1, where η is the tidal amplitude and $K_s$ is the Manning-Strickler friction coefficient.

Note that the friction number reflects the non-linear effect of the varying depth (Savenije, 2012). The tidal hydrodynamics analytical solution can be obtained by solving a set of four analytical equations, i.e., the phase lag equation, the scaling equation, the damping equation and the celerity equation (Cai et al., 2013). In Table 2, we present these equations for the general case as well as the special case of the ideal estuary (δ = 0).

### 1.2. Coupled model for salt intrusion

Since tidal dynamics in convergent alluvial estuaries can be reproduced reasonably well by one-dimensional analytical solutions, in principle the output of such model can be used to predict the longitudinal tidal excursion $E^*$ (Cai et al., 2015a): (i.e. $E^*$ is the dimensionless tidal excursion scaled by the frictionless tidal wave length), defined as:

$$E^* = E.\omega / c_0 = 2.v / c_0 \quad ; \quad v = r_s c_0 \mu \omega \eta / h \tag{11}$$

Where: $v$ is the velocity amplitude and ω is the tidal frequency, $c_0$ is the classical wave celerity of a frictionless progressive wave defined as:

$$c_0 = \sqrt{gh / r_s} \tag{12}$$

In which g is the acceleration due to gravity and $r_s$ the storage width ratio (e.g., Savenije et al., 2008).

On other hand, Van der Burgh (1972) assumed that the salinity curves for the HWS and LWS situations can be obtained by applying a horizontal translation over half the tidal excursion in the landward and seaward direction from TA situation and subsequently was demonstrated by Savenije (1986, 1989, 2005, 2012). Thus Eq. (6) can be used to describe the two situations of:

$$S^{*HWS}(x^*) = S^{*TA}(x^* + E^* / 2), \qquad S^{*LWS}(x^*) = S^{*TA}(x^* - E^* / 2) \tag{13}$$

Here the asterisk denotes a dimensionless variable.



The proposed analytical model by Cai et al., (2012a) for tidal hydrodynamics can be used to predict a variable velocity

amplitude $v$ (and hence tidal excursion $E^*$) for given tidal amplitude at the seaward boundary, estuary shape and friction.

## 1.3. Numerical modeling

### 1.3.1. Hydrodynamic model

Salinity distribution is influenced by the hydrodynamic regime, which in turn depends highly on the river estuary

morphology. In the hydrodynamics module, HEC-RAS solves the following one-dimensional equations of continuity and

momentum, known as the Saint-Venant equations (Brunner, 2010):

$$\frac{\partial Q}{\partial x} + \frac{\partial A}{\partial t} - q_l = 0 \tag{14}$$

$$\frac{\partial Q}{\partial t} + \frac{\partial (Q^2/A)}{\partial x} + g.A.\frac{\partial h}{\partial x} = g.A.\left\{ \left[ \frac{nQ}{AR^{2/3}} \right]^2 - \beta_0 \right\} \tag{15}$$

where Q is the discharge ($m^3$/s), A is the cross sectional area ($m^2$), x is the distance along the channel (m), t is the time (s), ql

is the lateral inflow per unit length ($m^2$/s), g is the acceleration due to gravity (m/$s^2$), h is the flow depth (m), $\beta_0$ (-) is the

bottom slope, n is the Manning's roughness coefficient ($m^{-1/3}$/s) and R is the hydrodynamic radius (m).

Manning coefficient used in the momentum equation is evaluated initially by the empirical formula equation (16) proposed

by Cowan (1956) and Chow (1973):

$$n = (n_0 + n_1 + n_2 + n_3 + n_4).m_5 \tag{16}$$

where $n_0$ is a basic, n is the value for a straight, uniform, smooth channel, $n_1$ is the adjustment for the effect of surface

irregularity; $n_2$ is the adjustment for the effect of variation in shape and size of the channel cross section; $n_3$ is the adjustment

for obstruction; n4 is the adjustment for vegetation; and $m_5$ is a correction factor for meandering channels.

The equations (14) and (15) are solved using the well known four-point implicit box finite difference scheme (Brunner,

2010).



This numerical scheme has been shown to be completely non-dissipative but marginally stable when run in a semi-implicit form, which corresponds to weighting factor (θ) of 0.5 for the unsteady solution. This value represents a half weighting explicit to the previous time step's known solution, and a half weighting implicit to the current time step's unknown solution. However, practically speaking, due to its marginal stability for the semi-implicit formulation, a θ weighting factor of 0.6 or more is necessary, since the scheme is diffusive only at values of θ greater than 0.5. In HEC-RAS, the default value of θ is 1. However, the user can specify any value between 0.6 to1 (Billah et al., 2015).

### 1.3.2. Mass transport model

In the advection-dispersion module, the basic equation is the one-dimensional advection-dispersion one of a conservative constituent (Brunner, 2010):

$$\frac{\partial(A.C)}{\partial t} = \frac{\partial}{\partial x}\left[D_x.A\frac{\partial C}{\partial x}\right] - \frac{\partial(Q.C)}{\partial x} \tag{17}$$

Where C is the salinity concentration (g/l), A is the cross-sectional area of the river ($m^2$), Q is the discharge ($m^3$/s) and $D_x$ is the longitudinal dispersion coefficient ($m^2$/s). This module requires output from the hydrodynamics module in terms of discharge, water level, cross-sectional area and hydraulic radius. The advection-dispersion equation is solved using the ULTIMATE QUICKEST explicit upwind scheme (Brunner, 2010). The resultant finite difference solution for equation (17) is as follows:

$$V^{n+1}C^{n+1} = V^n C^n + \Delta t\left(Q_{up}C_{up}^* - Q_{dn}C_{up}^* + D_{dn}.A_{dn}\left.\frac{\partial C^*}{\partial x}\right|_{dn} - D_{up}.A_{up}\left.\frac{\partial C^*}{\partial x}\right|_{up}\right) \tag{18}$$

where $C_{n+1}$is the concentration of a constituent at present time step ($g/m^3$), $C_n$ is the concentration of a constituent at previous time step ($g/m^3$), $C_{up}^*$ is the QUICKEST concentration of a constituent at upstream ($g/m^3$), $(\partial C^*/\partial x)_{up}$ is the QUICKEST derivative of a constituent at upstream ($g/m^4$), $D_{up}$ is the upstream dispersion coefficient ($m^2$/s), $V_{n+1}$: volume of the water quality cell at present time step ($m^3$), $V_n$ is the volume of the water quality cell at previous time step ($m^3$), $Q_{up}$ is the upstream flow ($m^3$/s), $A_{up}$ is the upstream cross-sectional area ($m^2$).



Inputs of the transport model are initial and boundary salinity concentrations and the dispersion coefficient (parameter D in equation (17)). The dispersion coefficient was estimated using Kashefipour and Falconer (2002) formula:

$$D = 10.612 . h . U \left( \frac{U}{U^*} \right) \qquad (19)$$

Where $U^*$ is the shear velocity, U is the cross-sectional average velocity, h is the depth of flow.

## 3. Overview of the Sebou estuary

The Sebou is the largest Moroccan river, draining approximately 40.000 km$^2$, stretching about 614 km from its source in the Middle Atlas Mountains to the Atlantic Ocean, which represents 6% of Morocco's total land area (Fig. 3). Kenitra harbor, about 17 km from the ocean, has commercial traffic, while Mehdia harbor at only 2 km from the mouth is busy with fishing activities. The flow regime at the level of the Sebou estuary is marked by considerable seasonal and inter-annual variations. It is under the influence of the tide regime and under the control of many dams (Igouzal and Maslouhi, 2005; Igouzal et al., 2005). During low flow periods, hydrodynamic regime is controlled by the Lalla Aïcha dam situated 62 km upstream. This dam has been constructed to preserve water for agricultural pumping stations and to avoid that salty waters rise towards these stations. Before the dam construction, excessive salinity reached up to 85 km upstream (Combe, 1969). Fig. 4 shows the flow release by Lalla Aicha dam from 01-01-2014 to 01-01-2016.

The tidal height varies from 0.9 to 3.10 m depending on the condition of the tide and the average flow is about 200 m$^3$/s at the river mouth Combe (1966). In addition, the tide near the estuary mouth is mainly semi-diurnal with a 44820-s tidal cycle (Haddout et al., 2014) and is a meso/micro-tidal estuary. The topography of the Sebou estuary is presented in Fig. 5. Fig. 5 shows the shapes of cross-sectional area, channel width, and depth during neap-spring and total lunar eclipse tides. The cross-sectional area and width are plotted directly from bathymetric data, and the water depth represents the ratio of these two geometric values.

The entire estuary can be divided into two reaches with different convergence length for cross-sectional area, width, and depth. The first inflection point is located at x =5km (between the river mouth (Mehdia) and Kenitra). The second inflection point is located at x=35km (between Oulad Salma and M'Rabeh), where the convergence length switches and the estuary becomes more riverine. Geometrical characteristics along the Sebou estuary are summarized in Table 3, with the cross-sectional area; width and depth are well described by exponential functions (Fig.5).



The convergence length is shorter in the seaward reaches (x = 0-5km and x=5-35km) where the tidal influence is dominant, regardless of the river flow upstream Lalla Aïcha dam. In the landward reach (x=35-62km), the influence of river flow becomes important and the convergence length much greater. In addition, we see a gradual depth increase along the estuary in the seaward reaches. Upstream the second inflection point, the depth gradually increases, while the cross-sectional area and width remains roughly constant.

## 4. Field data interpretation

In this study, five locations (Mehdia, Oulad Berjel, Kenitra, Oulad Salma, and M'Rabeh) along the Sebou estuary are chosen for salt measurements (Fig. 6) during 12 hours of 28$^{th}$ September 2015 (Super-moon day). A W-P600 conductivity meter is used in each location. The salinity can be expressed in parts per thousand (ppt or g/l) and the average value in the ocean is 35g/l. The period of measurements included LWS, TA and HWS situations (HWS and LWS correspond to periods when water velocity changes its directions and becomes nearly null).

The maximum and minimum salinity curves at HWS and LWS were thus observed, representing the envelopes of the salinity variation during tidal cycle. Fig. 6 shows vertically averaged salinity concentration measurements conducted at the five locations, during 12-h with 12-min interval. A maximum salinity concentration is recorded during high tides with 35.5 g/l at the river mouth, 32.7g/l at Oulad Berjel, 30g/l at Kenitra, 1.2g/l at Oulad Salma and 0.9g/l at M'Rabeh. A minimum salinity concentration is recorded during low tides with 17.5g/l at the river mouth and less than 1g/l in the other four locations.

On other hand, figure 7 shows vertically averaged salinity concentration measurements at normal situation during (spring tide) in three locations (Oulad Berjel, Kenitra, Oulad Salma). A maximum salinity concentration is recorded during high tides with 28.5g/l at Oulad Berjel, 19.8g/l at Kenitra and 0.9g/l at Oulad Salma. A minimum salinity concentration is recorded during low tides with 5g/l at Oulad Berjel and less than 3g/l in the other two locations.

The mixing mechanisms in estuaries are guided by tidal dynamics, the dispersion mechanisms and the amount of fresh water discharge from the river estuary. Dispersion includes longitudinal mixing, that takes place by mass travelling in streamlines at different velocities (Nylén and Ramel, 2012) and vertical mixing. Earlier measurements on Sebou estuary (Haddout et al., 2015b) has shown vertical salinity and temperature stratification, essentially for locations near the estuary mouth (Mehdia, Oulad Berjel, Kenitra), classifying Sebou estuary as a partially mixed river. On other hand, Fig. 8 shows water level measurements at Kenitra location from 27-09-2015 to 29-09-2015.





The water level drops in the low tide and then rises and peaks with the high tide to 4 m at total eclipse day (less than 3.2 m in normal situation). This indicates the total lunar eclipse influence on the estuary hydrodynamic regime.

## 5. Results and analysis

In this paper, the analytical (tidal propagation and salt intrusion) and numerical models introduced in the previous sections are applied to the Sebou estuary to evaluate the total lunar eclipse effect on salinity distribution.

### 1.4. Analytical hydrodynamic hybrid and salt intrusion method application

To turn the steady state model into a predictive model, semi-empirical relations (equation 6) are required that relate the two optimization parameters K and $D_1$ to hydrodynamic and geometrical bulk parameters. These bulk parameters are dimensionless numbers composed of geometrical (a, b, $A_0$, $B_0$, $h_0$), hydrological ($Q_f$) and hydraulic (H, E, C, $\upsilon$, $\eta$) parameters that influence the process of mixing and advection. An analytical hydrodynamic model for tidal wave propagation is used to reproduce the main tidal dynamics along the estuary axis and subsequently for predicting the main parameters (Van der

Burgh's coefficient K, dispersion coefficient $D_1$ and tidal excursion E) that influence the salt intrusion process during total lunar eclipse day of the Sebou estuary.

Field measurements of salt intrusion along the Sebou estuary axis, which were conducted at the 07[th]-18[th]-27[th] May 2015 (covering a spring-neap cycle) and at the 28[th] total lunar eclipse day, are used to test the predictive method. Each tidal cycle

consists of two HWS and two LWS salinity observations, which corresponds to the tidal wave periods. The hybrid (hydrodynamic) model presented in Sect.1 was calibrated during (07[th]-18[th]-27[th] May 2015) and at total lunar eclipse day. The calibrated parameters, the Manning-Strickler friction coefficient $K_s$ and the storage width ratio $r_s$, are presented in Table 4. It is worth noting that the storage width ratio $r_s$ is different in the seaward reaches for various tidal situations. It is important to point out that the model uses a variable depth in order to account for along-channel variations of the estuarine sections. Fig.9

shows the longitudinal computations of tidal amplitude and travelling time along the Sebou estuary for the selected periods (representing the spring tide, the moderate tide, the neap tide, and total lunar eclipse day). The agreement between analytically computed and observed tidal amplitude and travelling time for HW and LW is good. On other hand, these results shows, the difference between eclipse day and spring-neap situation is very remarkable. Additionally, a velocity amplitude and dumping number at eclipse day are show in the same figure, which suggests that the hybrid model proposed by Cai et

al., (2012a, 2014a) can well reproduce the tidal dynamics and velocity amplitude with a significant range of dam discharges.



Based on the hydrodynamic parameters (e.g., tidal amplitude and velocity amplitude) from hybrid (hydrodynamic) model, it is possible to estimate the main parameters that determine the salinity intrusion from predictive Eqs. (8)-(9) at the inflection point. In most estuaries, there usually exists an inflection point near the mouth, where the geometry changes (e.g., Gisen et

5  al., 2015b). On other hand, the Van Der Burgh's (K) and dispersion ($D_1$) coefficients are initially estimated by equations 8 and 9. However, due to the large uncertainty of these predictive equations, the K and $D_1$ estimations should be refined on the basis of salinity measurements. The optimization process of K and $D_1$ has been carried out using the Levenberg-Marquardt non-linear minimization method (Marquardt, 1963). In this method, the following objective function $\phi$ is minimized during the parameters optimization process.

$$\min \phi((K, D_0), S) = \frac{\sum_{i=1}^{m} (S_i - S_i^+(K, D_1))^2}{m . \sigma_S^2} \qquad (20)$$

Where $S_i$ and $S_i^+$ are the measured and predicted salinity, $\sigma_S^2$ are variance of the measured salinity, m is number of observation.

The Levenberg-Marquardt algorithm adaptively varies the parameter updates between the gradient descent update and the Gauss-Newton update.

Because the predictive uncertainty of these equations, an optimization is applied to K and $D_1$ so as to obtain the best fit between computed salt-intrusion curve and in-situ data (salinity). Values of the optimized parameters are summarized in Table 5 (at HWS), where the dispersion coefficient at the estuary mouth $D_0$ can be obtained by substituting $D_1$; $x_1$ and K into Eq. (4). It can be shown that the estimated values of the parameters $D_1$ and K are very close to their reference value (equations 8 and 9). At each tidal condition (HWS, LWS, and TA) the optimized values of $D_1$ are greater that initial

estimated values, whereas the optimized values of the coefficient K have remained constant and equal to 0.20 for all tidal conditions at total lunar eclipse day. Also, in normal situations (spring-neap cycle) Haddout et al, (2015b) founded a value of K=0.15 which is relatively small compared to the value of K=0.20 during eclipse day.

We attribute this difference to the dominance of tidal mixing in the Sebou estuary for the eclipse period. Because the

optimized value of K remained constant, fitted salinity curves were more sensitive to the dispersion coefficient D at HWS, LWS, and TA (Haddout et al., 2015b). Results of the axial salinity analysis at HWS, TA and LWS are plotted in Fig. 10. On the whole, it can be said that the analytical salt intrusion model performs well in representing the salinity distribution in the Sebou estuary (surveyed on 28th September 2015).





Additionally, salt-intrusion exhibit three distinct tendencies: a dome shape at HWS and TA, a recession and bell shapes at LWS (see Appendix A). These three salinity shapes were observed during the eclipse day.

On other hand, in a positive estuary, the salinity gradient is always negative due to the decreasing salinity in landward direction as it is the case for the Sebou river estuary (Fig. 11a). The dome shaped intrusion curves have a negative curvature in the seaward part of the estuary. As water level increases, the position of the maximum salinity gradient moves towards to the estuary downstream, after which it has a dome shape with monotonous increasing of the salinity gradient . In addition, Fig. 11b shows curves of second derivative of salinity gradient.

Additionally, the scatter plot of the computed vs optimized result for dispersion coefficient at High Water Slack is shown in Fig.12.

*The predictive model compared to other methods*

This salinity intrusion model has been applied in different estuaries all over the world and by several methods (e.g.Van den
Burgh, 1972; Rigter, 1973; Fischer, 1974; Van Os and Abraham, 1990 and Savenije, (1993a, 2005) see Appendix B).

In Fig. 13, the observed vs computed intrusion lengths are plotted together with data specific to Sebou estuary. As one can see in the same figure, the result of Savenije model is considered most accurate.

**1.5. Numerical modelling of salinity distribution**

**1.5.1. Hydrodynamic model**

The hydrodynamic regime was first studied and modeled in HEC-RAS. Outputs from the hydrodynamic model (velocity and water level evolution) were used in the salt transport study. The final resolution of hydrodynamic model equations (14 and 15) requires spatial discretization of the study area. The river reach (62 km) was discretized into 203 grids with a length
varying between 58 and 996 m (Haddout et al., 2015a). Data on cross sectional (bathymetry) areas from the ANP (National Agency of Ports) and other sources were used. The upstream boundary (at the Lalla Aïcha dam) was given values of discharge as a function of time from 27-09-2015 to 29-09-2015. Also, the downstream boundary (at the mouth) was given values of the water level as a function of time.

The factor $n_0$ equation (16) is evaluated from granulometric measurements that were carried out from upstream to
30 downstream in the studied reach. The others coefficients were evaluated from observations of the river in aerial photos, from the cross- sectional areas and available photos, and from field visits.





The hydrodynamic model has been calibrated and valided using data from 27-09-2015 to 29-09-2015. The calibrated parameter is Manning's roughness in the estuary. The calibration and validation are performed using the water level data at kenitra location. The 27-09-2015 day has been used for calibration. The roughness coefficients were adjusted by a trial and error approach until the simulated and observed water levels were satisfactory.

Fig. 14a shows the comparison of the simulated water level at Kenitra location with the observed data where the water levels are measured based on the datum of the National Agency of Ports (ANP). For the model validation, water levels during 28-09-2015 and 29-09-2015 have been used. Fig. 14b shows good correspondence between the observed and simulated water levels at Kenitra location.

### 1.5.2. Salt transport model

The salinity model has been calibrated by systematically adjusting the values of a selected system parameter to achieve an acceptable match between the measured salinity and the corresponding values computed by the one dimensional advection dispersion model. The calibration parameter is the dispersion coefficient in the river which is adjusted by trial and error. Fig. 15 shows the comparison of the observed and simulated salinity at three locations (Oulad Berejel, Kenitra, Oulad salma) during the total eclipse day.

In the calibration procedure, the dispersion coefficient was modified to the same degree along the studied reach because we assumed that the sources of errors involved in its evaluation are identical for all the grids. The calibrated value of coefficient 'D' ranges from 500 to 900 $m^2/s$ along the river. The results show that the simulated salinity concentration follows a similar trend to the observed data.

## 6. Models Performance Verification

The statistical indicators used for evaluating the performance of the numerical and analytical models are: root mean squared error (RMSE); mean absolute error (ABSERR); the Nash Sutcliffe modelling efficiency index (EF); the goodness-of-fit ($R^2$) and the % of deviation from observed streamflow (PBIAS). The statistical parameters were defined as follows (Moriasi et al., 2007; Stehr et al., 2008; Conversa et al., 2015):

$$RMSE = \sqrt{\left[ \frac{\sum_{i=1}^{n}(O_{meas,i} - S_{pred,i})^2}{N} \right]} \tag{21}$$





$$ABSERR = \left[ \frac{\sum_{i=1}^{n}(O_{meas,i} - S_{perd,i})}{N} \right] \qquad (22)$$

$$EF = 1 - \left[ \frac{\sum_{i=1}^{n}(O_{meas,i} - S_{perd,i})^2}{\sum_{i=1}^{n}(O_{meas,i} - \overline{O}_{meas})^2} \right] \qquad (23$$

$$R^2 = \left[ \frac{\sum_{i=1}^{n}(O_{meas,i} - \overline{O}_{meas})(S_{pred,i} - \overline{S}_{pred})}{(\sum_{i=1}^{n}(O_{meas,i} - \overline{O}_{meas})^2)^{1/2}(\sum_{i=1}^{n}(S_{pred,i} - \overline{S}_{pred})^2)^{1/2}} \right]^2 \qquad (24)$$

$$PBIAS = \left[ \frac{\sum_{i=1}^{n}(O_{meas,i} - S_{perd,i})}{\sum_{i=1}^{n}(O_{meas,i})} .100 \right] \qquad (25)$$

Where $O_{meas,i}$ is the observed value and $S_{pred,i}$ the computed value of salinity or water level. $\overline{O}_{meas}$ is the mean observed salinity or water level data and $\overline{S}_{pred}$ is the mean computed salinity or water level.

10   The closer the values of RMSE and ABSERR to zero, and the $R^2$ to unity, the better the model performance is evaluated (Abu El-Nasr et al., 2005). For Percent bias (PBIAS) measures the average tendency PBIAS, expressed as a percentage, of the simulated data to be larger or smaller than their observed counterparts (Gupta et al., 1999). The optimal value of PBIAS is 0, with low-magnitude values indicating accurate model simulation (Moriasi et al., 2007). Positive values indicate model underestimation bias and negative values indicate model overestimation bias (Gupta et al., 1999). The Nash-Sutcliffe

15   efficiency (EF) (Nash and Sutcliffe, 1970) is a normalized statistic that determines the relative magnitude of the residual variance (noise) compared to the measured data variance (information). EF ranges between $-\infty$ and 1 (1 inclusive), with EF = 1, the closer the model EF efficiency is to 1, the more accurate is the model. Values between 0 and 1 are generally viewed as acceptable levels of performance, whereas values $\leq 0$ indicate unacceptable performance (Moriasi et al., 2007).





The indicators of hydrodynamic-salinity intrusion model are summarized in Tables 6 and 7. The two models are the EF and $R^2$ coefficients are very near to unity. This result demonstrates the good performance of the analytical model. Also, this shows that the proposed coupled analytical model by Cai et al., (2015a) is applicable and useful.

The statistical performances of the numerical model uses water level for comparison. Values of statistical parameters indicated in Table 8 show good correlation model calculations and measurements during calibration and validation. These indicate that the model can estimate the water level at Kenitra fairly well.

The statistical indicators for transport model are summarized in Table 9. The results show that the computed salinity concentration follows observed data, which suggest that the presented mass transport model is a reasonably efficient tool for predicting the impact of total lunar eclipse on salt intrusion in alluvial estuaries.

## 7. Total lunar eclipse impact

The impact of eclipse on river hydrodynamics is mainly caused by a moon closest to the earth, causing strongest gravitational pull. The maximum salinity at high water along the Sebou estuary has been described in section 4. Total lunar eclipse impact on the maximum salinity at different locations compared to the normal situation is given in Table 10. The results clearly show that the lunar eclipse impact on salinity intrusion is highly significant.

On other hand, Fig. 16 shows the profiles of salinity during total lunar eclipse compared to the spring-neap tides. It appears that the salt intrusion curve computed in the neap-spring tides are recession-type, while it becomes a dome-type shape at eclipse day. According to Nguyen et al. (2012), this is subjected to changes in the degree of convergence of the cross-sectional profile, and the effect of the mixing due to freshwater discharge (i.e. that increasing the tidal amplitude at the mouth tends to produce shorter convergence lengths of the cross-sectional area and width). The convergence or divergence of the channel can dramatically change the shape of the salt intrusion curve (Gay and O'Donnell, 2007; Cai et al., 2015b). In addition, Savenije (2005), shows that the recession-type curve occurs in narrow estuaries having a near-prismatic shape and high river discharge and dome-type shape which occurs in strong funnel-shaped estuaries (with a short convergence length) (see Appendix A). At eclipse day, when the channel converges strongly, the mixed water retains relatively higher salinity from the estuary mouth. However, salinity profiles under all spring-neap tides show a gradual decrease from the mouth to the upstream reach.



Additionally, it can be shown that the part of the Sebou estuary that is affected by the total lunar eclipse is from 20 to 25 km upstream of the river mouth. A water level rise as showed above during this exceptional event moves the excessive salinity
5 (25g/l) until 20 km upstream. On other hand, water level rise causes large augmentation of salinity in the mesohaline zone of the Sebou estuary. Also, if we considerer Kenitra location, an increase of water level for 0.8m causes an increase of 9.4 g/l in salinity witch correspond to a salinity augmentation of 11.75 g/l per metre of increased water level.  At the Oulad salma drinking water station salinity increased to a value of 1.2 g/l that exceeds the limit value of 500 mg/l recommended by the World Health Organization (WHO) for drinking water.

Computations using hybrid (hydrodynamic) and salt intrusion models during total lunar eclipse required the recalculation of the geometric parameters of the estuary i.e., cross-sectional areas A0, convergence lengths a and the optimization of the dispersion coefficients D. Geometry is one of the most important parameters in the hydrodynamic and salt intrusion models. It affects the character of the salinity distribution and appears prominently in the shape of salt intrusion curves during
extreme events. Computed results reveal that variations in the sensitivity of these parameters are likely to depend on changes in geometric characteristics.

## 8. Conclusions

The purpose of this paper was to study the impact of the total lunar eclipse of 28th September 2015 on salt intrusion in Sebou river estuary. It is, to our knowledge, the first time that this infrequent phenomenon is studied in terms of its influence
on water quality. Field measurements showed a change of the salinity profiles form along the estuary axis and a notable water level rise, compared with normal situations studied in our earlier works. In addition, results show that the average salt content increased in the reach between 0-25 km, as a result of water volume rise at mouth. An hybrid proposed bu Cai et al. (2014) coupled to analytical salt intrusion model in alluvial estuaries (Savenije, 2005) and a numerical model (HEC-RAS) have been applied in the Sebou river estuary. Calculations during lunar eclipse using the coupled hybrid-salt intrusion model
required the recalculation of the geometric parameters of the estuary i.e., cross-sectional areas $A_0$, convergence lengths a. A good fit was obtained between computed and measured salinity during this extreme event. These models reproduce very well the salinity rise. Statistical indicators show that these models fit adequately salinity observations in the Sebou estuary.





A comparison between the two applied models is not the objective of this study since each one can be applied for specific management purposes. The analytical models are helpful for situations where a quick longitudinal salinity profiles is needed. On other hand, the numerical 1-D model is powerful where a temporal salinity variation is carried out in a specific location, but it needs more data and time for its implementation. Hence, these tools can be very helpful for water managers and engineering to make preliminary estimates on the salt intrusion along the estuary axis even during extreme events. These extreme events can concern similar total lunar eclipse, see level rise due to climate change, a sea tsunami.

Finally, the impact of extreme events on the water quality of Sebou estuary should be considered by managers. Rapid interventions, based on the predictions of our mathematical models can thus be taken. These interventions may involve the pumping station closure for example.

# Acknowledgement

The authors would like to express their gratitude to: H. Qanza, M. Hachimi, O. khabali and O. El Mountassir for the efforts in the field measurements during the total lunar eclipse day. The authors would also like to acknowledge the technicians the water services of Kenitra; and the engineers of the National Agency of Ports for their availability and collaboration.

**Appendix A. Types of salt intrusion and shape of salt intrusion curves**

Salinity distribution is a veritable fingerprint of each estuary and in direct relation to both its geometric form and hydrology. For partially mixed and well-mixed estuaries, a number of designations are used to classify salinity profiles into three types depending on their shape. The following types are distinguished (Savenije, 2005, 2012) (see Fig. 17):

-*Recession shape*, which occurs in narrow estuaries with a near-prismatic shape and a high river discharge (Savenije, 2005, 2012).

- *Bell shape,* which occurs in estuaries that have a trumpet shape, i.e. a long convergence length in the upstream part, but a short convergence length near the mouth (Savenije, 2005, 2012).

- *Dome shape,* which occurs in strong funnel-shaped estuaries (with a short convergence length) (Savenije, 2005, 2012).





**Appendix B. Empirical models** (Savenije, 2003a):

-Rigter (1973):

$$L^{LWS} = 1.5\,\pi\,\frac{h_0}{f}\,(F_D^{-1}.N^{-1} - 1.7) = 4.7\pi\,\frac{h_0}{f}\,F_D^{-1.0}.N^{-1} \qquad (26)$$

-Fischer (1974):

$$L^{LWS} = 17.7\,\frac{h_0}{f^{0.625}}\,F_D^{-0.75}.N^{-0.25} \qquad (27)$$

-Van der Burgh (1972):

$$L^{TA} = -26\frac{h_0}{K}\,\frac{\sqrt{g.h_0}}{v_0}\,\frac{v_0}{u_0}.N^{0.5} = 26.\pi\frac{h_0}{K}.F^{-1.0}.N^{-0.5} \qquad (28)$$

-Van Os and Abraham (1990):

$$L^{LWS} = 4.4\frac{h_0}{f}\,F_D^{-1}.N^{-1} \qquad (29)$$

Where $h_0$ is the tidal average depth, $v_0$ is the maximum tidal velocity, $u_0$ is the velocity of freshwater, N is the Canter

Cremers' estuary number, K is the Van der Burgh's coefficient, f is the Darcy-Weisbach's coefficient, F is the Froude

number and $F_D$ is the densimetric Froude number.

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



**Tables:**

Table 1. Definition of dimensionless parameters.

| Dimensionless parameters | |
|---|---|
| Local variable | Dependent variable |
| Tidal amplitude $\zeta = \eta / h$ | Damping number $\delta = c_0 d\eta / (\eta \omega dx)$ |
| Estuary shape $\gamma = c_0 / (\omega a)$ | Velocity number $\mu = v / (r_s \zeta c_0) = vh / (r_s \eta c_0)$ |
| Friction number $\chi = r_s f c_0 \zeta / (\omega h)$ | Celerity number $\lambda = c_0 / c$ |
| River discharge $\varphi = U_r / v$ | Phase lag $\varepsilon = \pi / 2 - (\phi_Z - \phi_U)$ |

**Table 2.** Hybrid solution of tidal wave propagation in convergent estuaries (Cai et al., 2015b).

| Case | Phase lag $\tan(\varepsilon)$ | Scaling $\mu$ | Damping $\delta$ | Celerity $\lambda^2$ |
|---|---|---|---|---|
| General | $\lambda / (\gamma - \delta)$ | $\sin(\varepsilon) / \lambda = \cos(\varepsilon) / (\gamma - \delta)$ | $\gamma / 2 - 4\chi\mu / (9\pi\lambda) - \chi\mu^2 / 3$ | $1 - \delta(\gamma - \delta)$ |
| Ideal estuary | $1 / \gamma$ | $\sqrt{1 / (1 + \gamma^2)}$ | 0 | 1 |

**Table 3.** Geometric characteristics in the Sebou estuary.

| Reach [km] | Tidal conditions | $A_0$ [m²] | $B_0$ [m] | $h_0$ [m] | a [km] | b [km] | d [km] | T [s] |
|---|---|---|---|---|---|---|---|---|
| | Neap-Spring | 2300 | 285 | 5.50~5.8 | 5.24 | 4.40 | -27.45 | 44820 |
| 0~5 | Eclipse | 4900 | 512 | 6.83~7.2 | 2.84 | 2.01 | -6.88 | 44820 |
| | Neap-Spring | 1100 | 250 | 4.10~3.1 | 49 | 34 | -111 | 44820 |
| 5~35 | Eclipse | 3172 | 477 | 5.23 | 51 | 36 | -122.4 | 44820 |
| | Neap-Spring | 750 | 150 | 3.20~2 | 57 | 60 | 1140 | 44820 |
| 35~62 | Eclipse | 1700 | 316 | 4.24 | 58 | 60 | 1740 | 44820 |

**Table 4.** Calibrated parameters for the hydrodynamic model of the Sebou estuary.

| Reach [km] | Storage width ratio $r_s$ (-) | | | | Manning-Strickler friction $K_s$ (m$^{1/3}$.s$^{-1}$) |
|---|---|---|---|---|---|
| | Spring | Moderate | Neap | Eclipse day | |
| 0~5 | 1.20 | 1.30 | 1.30 | 1.01 | 67 |
| 5~35 | 1.20 | 1.30 | 1.30 | 1.01 | 65 |
| 35~62 | 1.40 | 1.40 | 1.40 | 1.30 | 41 |



**Table 5.** Salinity distribution data showing the salinity at the mouth ($S_0$), tidal excursion (E), Richardson number ($N_R$), dispersion coefficient at High Water Slack, Van Der Burgh's coefficient (K) and salt intrusion length at High Water Slack (L).

| Tidal conditions | $S_0$ [g/l] | E [km] | $N_R$ [-] | $D_1^{HWS}{}_{Computed}$ [m²/s] | $D_1^{HWS}{}_{Optimzed}$ [m²/s] | $K_{Computed}$ [-] | $K_{Optimzed}$ [-] | $L_{Comp}$ [km] | $L_{Obs}$ [km] |
|---|---|---|---|---|---|---|---|---|---|
| Total.Lunar. | 35.8 | 12.31 | 0.047 | 533.61 | 590.90 | 0.18 | 0.20 | 29.0 | 25.0 |
| eclipse (1st at HWS) | 35.8 | 13.00 | 0.039 | 529.23 | 602.76 | 0.18 | 0.20 | 28.6 | 26.0 |
| | 35.8 | 12.73 | 0.042 | 527.46 | 535.39 | 0.18 | 0.20 | 26.0 | 25.0 |
| Total.Lunar. | 35.8 | 08.63 | 0.140 | 688.20 | 700.31 | 0.18 | 0.20 | 25.0 | 27.0 |
| eclipse (2nd at HWS) | 35.8 | 06.63 | 0.300 | 742.59 | 792.02 | 0.18 | 0.20 | 25.6 | 27.0 |
| | 35.6 | 06.53 | 0.310 | 733.20 | 788.18 | 0.18 | 0.20 | 25.9 | 27.9 |
| Spring(at HWS) | 35.0 | 06.50 | 0.305 | 440.48 | 403.25 | 0.16 | 0.15 | 23.1 | 21.5 |
| Neap (at HWS) | 34.5 | 08.00 | 0.150 | 415.40 | 400.00 | 0.15 | 0.15 | 23.3 | 20.0 |

**Table 6.** Statistical indicators of analytical hydrodynamic model performance during eclipse day.

| Statistical indicators of hybrid model during eclipse day | RMSE [m, min] | ABSERR [m, min] | EF [-] | $R^2$ [-] | PBIAS [%] |
|---|---|---|---|---|---|
| Tidal amplitude (1st and 2nd) | 0.42~0.49 | 0.37~0.53 | 0.79~0.89 | 0.91~0.92 | 1.42~1.7 |
| Travel time at HW (1st and 2nd) | 0.50~0.65 | 0.32~0.44 | 0.88~0.94 | 0.89~0.93 | 1.01~1.1 |
| Travel time at LW (1st and 2nd) | 0.55~0.61 | 0.30~0.47 | 0.97~0.95 | 0.95~0.97 | 0.98~1.0 |

**Table7.** Statistical indicators of analytical salinity intrusion model performance at HWS during eclipse day.

| Statistical indicators of salinity model during eclipse day | RMSE [g/l] | ABSERR [g/l] | EF [-] | $R^2$ [-] | PBIAS [%] |
|---|---|---|---|---|---|
| | 0.52 | 0.27 | 0.82 | 0.89 | 2.82 |
| 1st(at HWS) | 0.73 | 0.63 | 0.86 | 0.90 | 2.60 |
| | 0.83 | 0.70 | 0.90 | 0.90 | 1.36 |
| | 0.48 | 0.72 | 0.94 | 0.93 | 1.73 |
| 2nd(at HWS) | 0.53 | 0.76 | 0.94 | 0.92 | 1.36 |
| | 0.73 | 0.81 | 0.94 | 0.93 | 2.30 |

**Table 8.** Statistical indicators of Hydrodynamic model performance in calibration and validation

| Statistical indicators of hydrodynamic model | RMSE [m] | ABSERR [m] | EF [-] | $R^2$ [-] | PBIAS [%] |
|---|---|---|---|---|---|
| Calibration (27-09-2015) | 0.34 | 0. 21 | 0.94 | 0.93 | 0.94 |
| Validation (from 28-09-2015 to 29-09-2015) | 0.66 | 0.59 | 0.90 | 0.89 | 1.01 |




**Table 9.**Statistical indicators of transport model performance in calibration.

| Statistical indicators of transport model in calibration | RMSE [g/l] | ABSERR [g/l] | EF [-] | $R^2$ [-] | PBIAS [%] |
|---|---|---|---|---|---|
| Oulad Berjel | 0.69 | 0.71 | 0.96 | 0.91 | 0.92 |
| Kenitra | 0.88 | 0.86 | 0.92 | 0.86 | 1.04 |
| Oulad salma | 0.92 | 0.89 | 0.92 | 0.84 | 1.12 |

5  **Table 10.**Comparaisons salinity variation at HWS in different locations

| Estuary locations | Normal situation-HWS salinity (g/l) | Total eclipse day-HWS salinity (g/l) |
|---|---|---|
| Oulad Berjel | 28.5 | 32.6 |
| Kenitra | 16.6 | 26.0 |
| Oulad salma | 00.7 | 01.2 |
| M'Rabeh | 00.6 | 00.8 |

**Figures:**

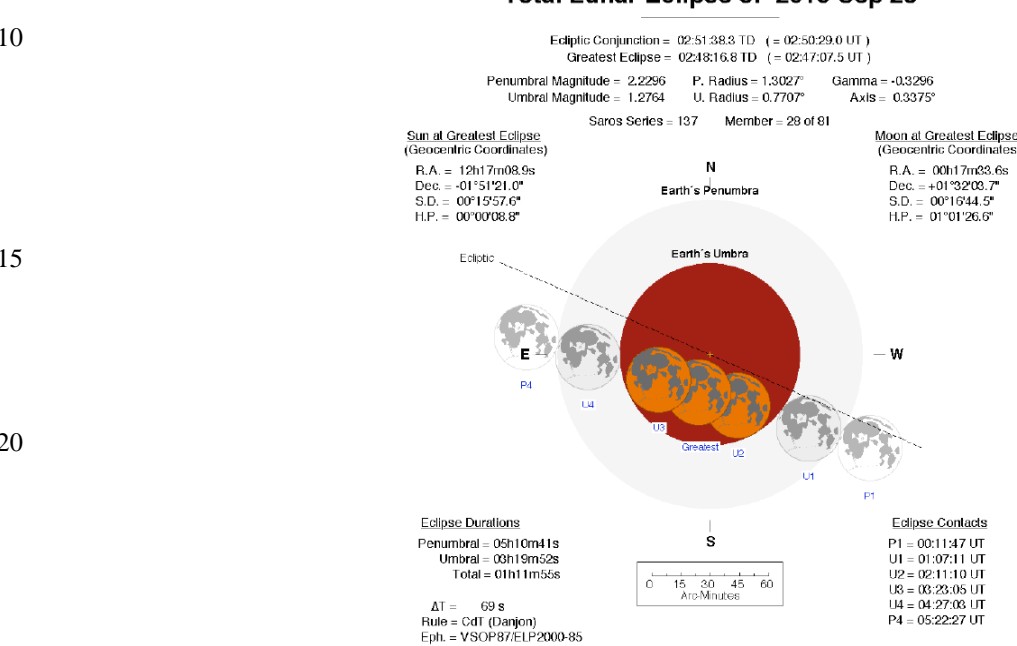

25  **Figure 1.**Total lunar Eclipse calculation by Fred Espenak, NASA/GSFC.





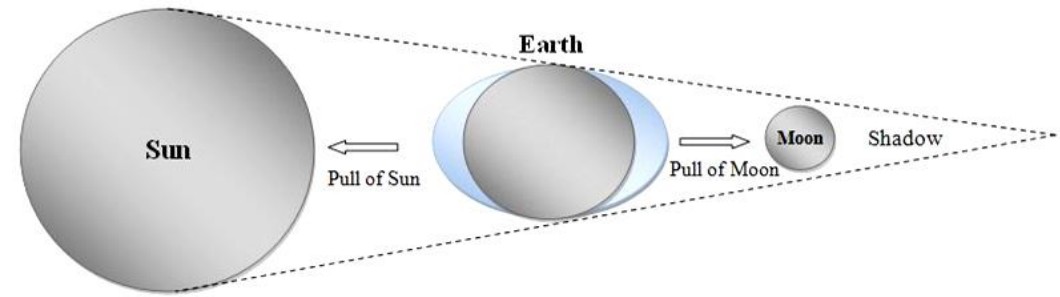

**Figure 2.**Illustrating the position of the moon (Total lunar eclipse).

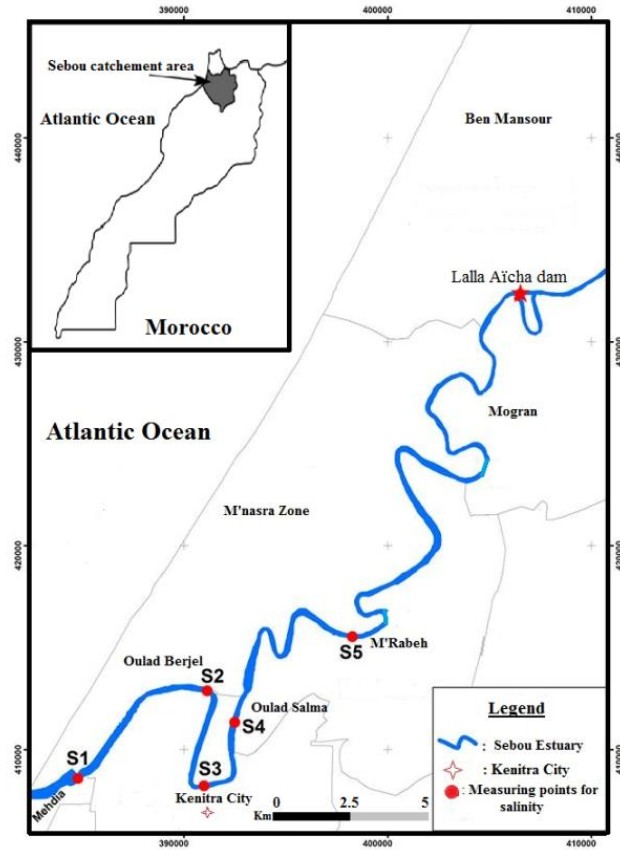

25         **Figure 3.**Study area and measurements sites in the Sebou river estuary.





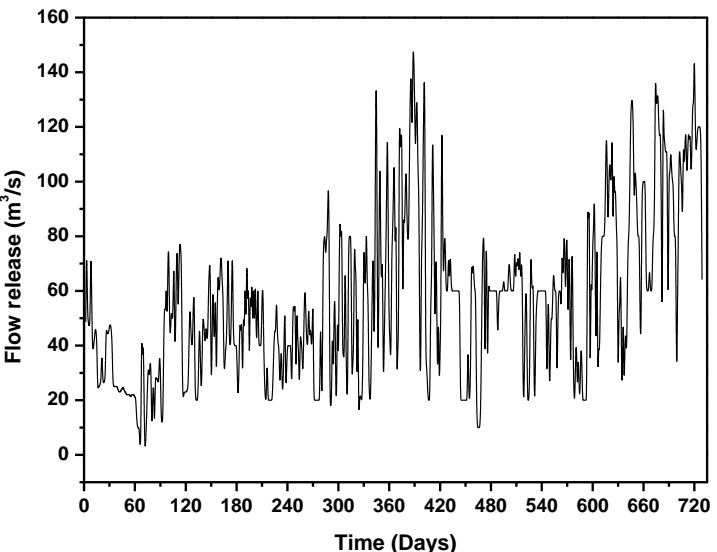

**Figure 4.** Flow release by Lalla Aicha dam (from 01-01-2014 to 01-01-2016).

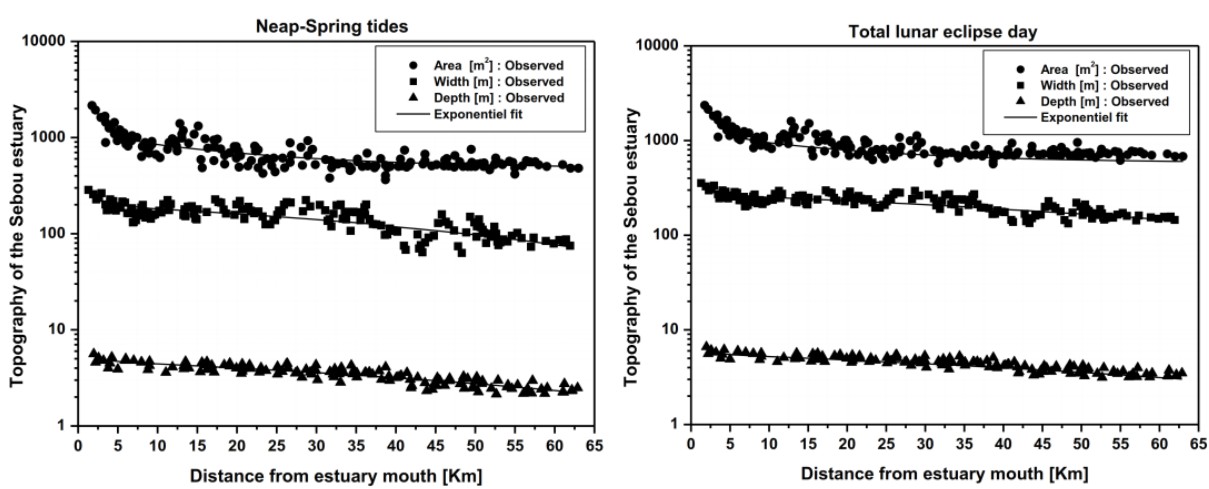

**Figure 5.** Geometry of the Sebou estuary, showing the cross-sectional area A (m²), the width B (m), and the estuary depth h

(m) during neap-spring and total lunar eclipse tides.





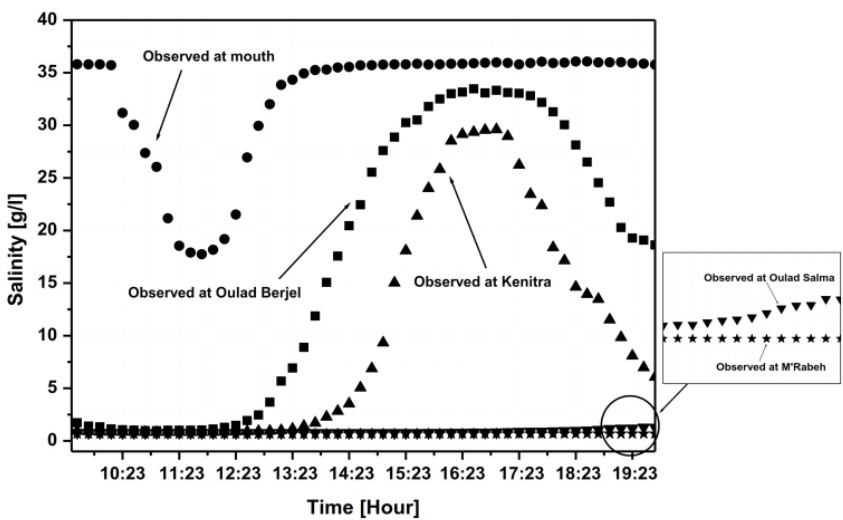

**Figure 6.**Salinity evolution in the Sebou river estuary during 12-h (Surveyed on 28 September 2015).

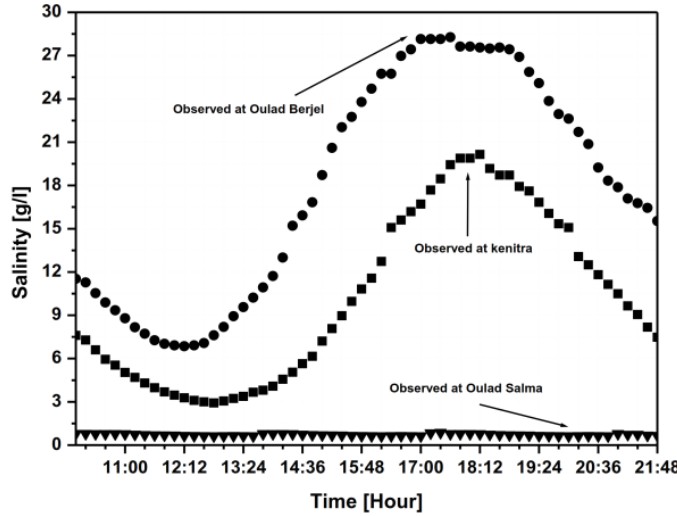

**Figure 7.**Salinity evolution in the Sebou river estuary during Spring tide (Surveyed on 11 February 2016).



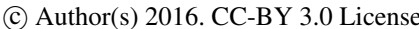

**Figure 8**.Water level measurements at Kenitra location from 27-09-2015 to 29-09-2015.







**Figure 9**.Analytically computed tidal amplitude, travelling time, velocity amplitude and damping number along the Sebou estuary at total lunar eclipse day (tidal amplitude, travelling time and velocity amplitude are compared against observations).





**Figure 10.**Observed and analytically computed longitudinal salinity distribution along the Sebou estuary (surveyed on 28[th] September 2015) during 1[st] cycle (a, b, c) at LWS, TA and HWS, and 2[nd] cycle (d,e,f) at LWS TA and HWS.



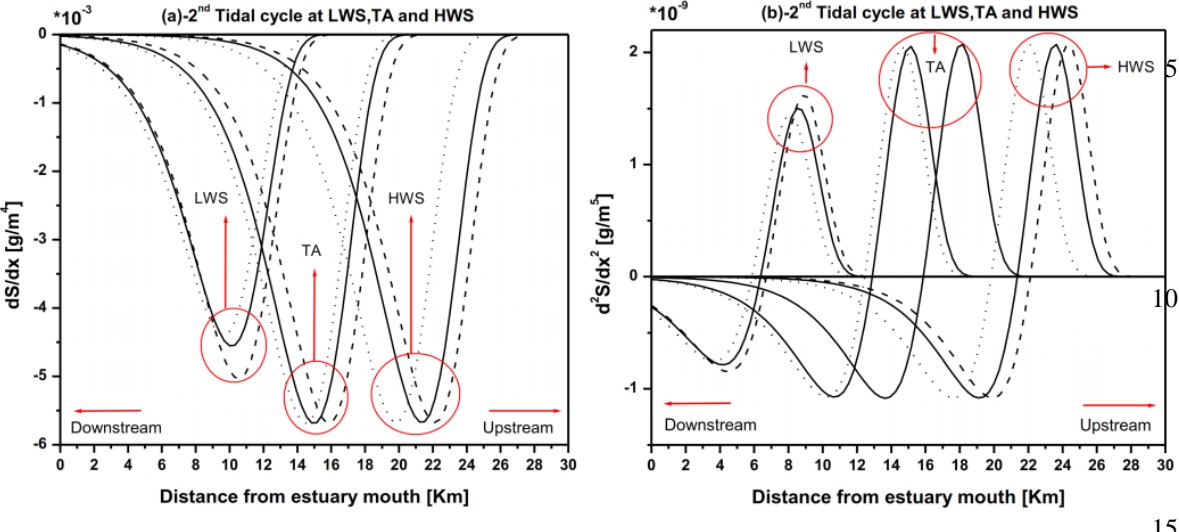

**Figure 11**.Longitudinal variation of salinity gradient (a) and curvature (second derivative) (b) along the Sebou estuary axis.

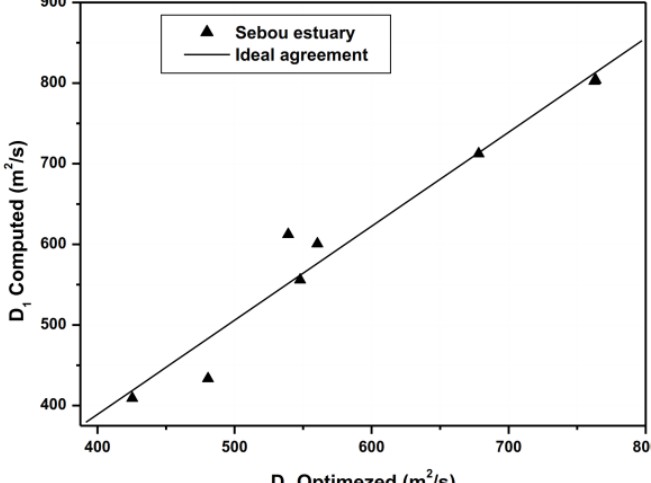

**Figure 12**.Comparison of computed and optimized dispersion $D_1$ coefficient at HWS in the spring-neap and eclipse day tides.



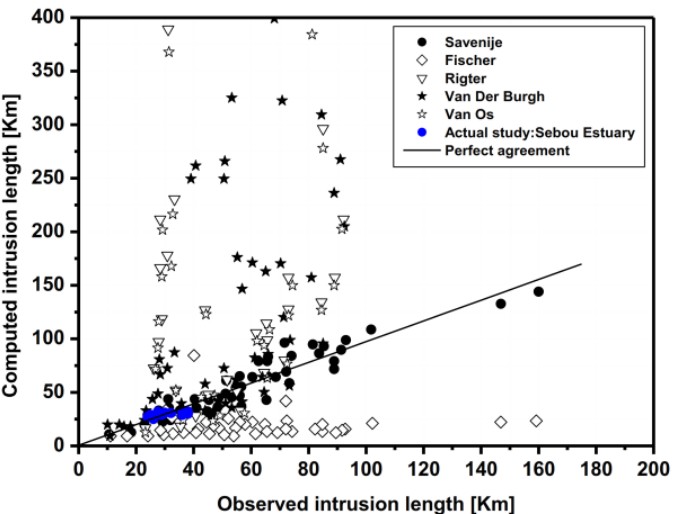

**Figure 13**.Comparison of different methods to compute the intrusion length at HWS and actual study (Sebou estuary)

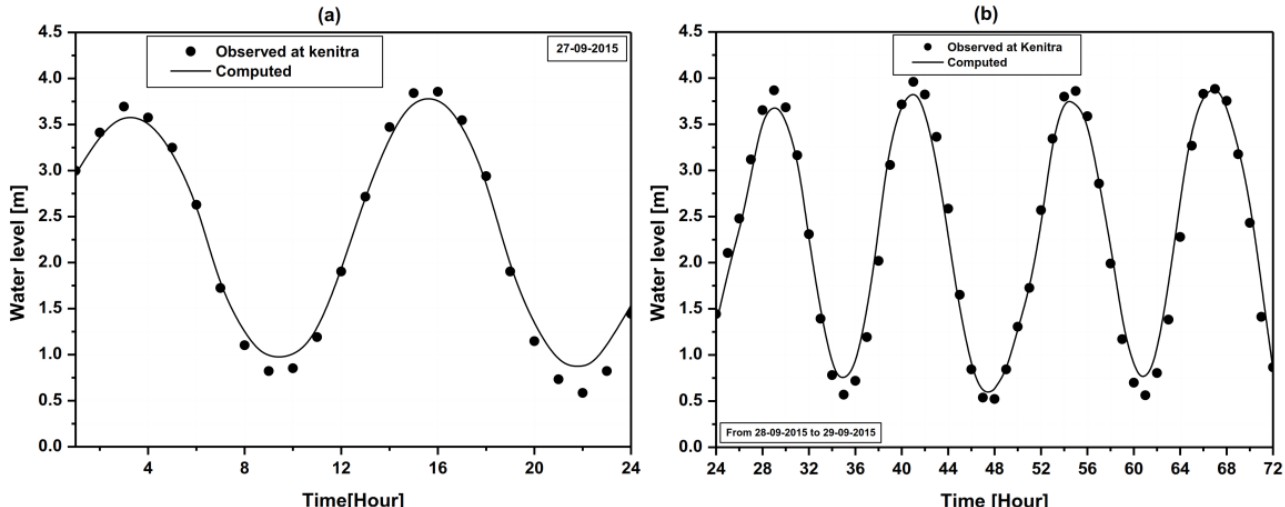

**Figure 14**.Water level comparisons at kenitra location in calibration (a) and validation (b).





**Figure 15**.Salinity comparisons at Oulad Berjel (a), Kenitra (b) and Oulad Salma (c) in calibration.





5 **Figure 16**.Salinity distributions in the Sebou estuary during spring tide, neap tide and total lunar eclipse day.

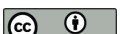



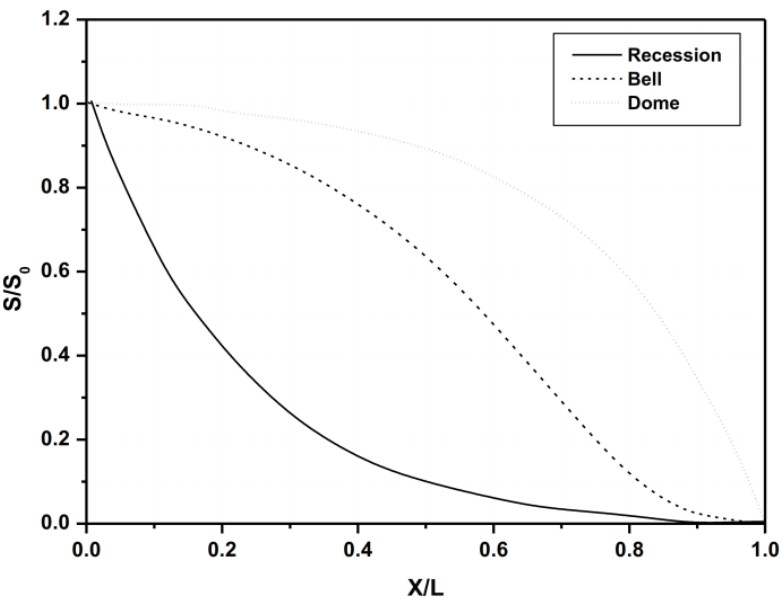

**Figure 17**.Three types of salt intrusion curves, in which L is the salt intrusion length, x is the distance from the mouth, S is the salinity at the mouth and $S_0$ is the salinity corresponding with the distance.