# Peer review of "Analytical and numerical study of the salinity intrusion in the Sebou river estuary (Morocco). Effect of the ‘Super Blood Moon’ (total lunar eclipse) of 2015"

_Hydrology and Earth System Sciences, 2016_

## Referee Comment (RC1) · Anonymous Referee #1 · 18 Jun 2016

This is a well-prepared and well-documented paper. Besides some minor editing that is required, the paper is well written. I presume the publisher will take care of the English editing and minor corrections. (just as a reminder, please make sure to replace dumping by damping in P.12, L29.

The case of analysing an extreme spring tide in an intensively used estuary is very relevant. It is definitely a cutting-edge case study. The study has been very well done, and it is based on a very detailed data set of intensive measurements. This makes this paper very interesting and relevant for HESS.

[Figure]

Although the analysis is clear in general, there is one thing that I don't understand, and which needs clarification and correction, as far as I'm concerned.

In Figure 10, and also in Table 5, we three moments in time for HWS, TA and LWS. This is strange, because at every cycle, there is only one moment of HWS, LWS and TA. Is this because the authors did not observe the moment of slack, but just derive it from the temporal observation of the salinity (as in Figure 6). But if that is so, than the maximum value corresponds to HWS, the minimum value to LWS and the time-average value to TA. There should not be multiple values. Also in Figure 10, I don't understand why the differences between the observations that are only 20 min apart are so large. In Figure 6 the maximum values of the salinity curves are rather flat. So the differences should not be large.

Maybe the authors determined the moment of HWS on the basis of the hydraulic model. But that would be wrong, since the hydraulic model may determine the moment of slack wrongly. The correct moments of slack, if not observed in the field should correspond with the maximum and minimum observed salinities.

Some minor comments:

If the hydraulic model is calibrated on the Roughness, then it is useless to present the composite equation for the Manning roughness (16).

Similarly, if the dynamic 1-D salt balance equation is calibrated on the Dispersion, then don't mention (19) in the paper. Moreover equation (19) is not at all appropriate for salt intrusion. It refers to rivers only. So eqs (16) and (19) should be removed from the paper.

Also use the same parameters throughout the paper. So if you use K_manning in one equation then don't use n in another.

Table 4: Interesting that you note that R_s is larger during neap. This was also observed earlier by Zhang et al., 2012.

Zhang, E., H. H. G. Savenije, S. L. ChenïiǰNX. H. Mao, 2012. An analytical solution for tidal propagation in the Yangtze Estuary, China. Hydrol. Earth Syst. Sci., 16, 3327-3339.

---

## Referee Comment (RC2) · Anonymous Referee #2 · 16 Aug 2016

This work takes the opportunity to capture the effects from an extreme event (super moon total eclipse). The rarity of this event, in terms of the tidal power, comes from two factors: the moon is at its closest distance to earth (super moon), and the exact line up of the sun, the moon and the Earth (total eclipse). Comparing both analytical and numerical models to field measures taken right during the event gives significance to this work. The results clearly show the power of relatively simple model (1D) and the effectiveness of previously developed theoretical and empirical methods for estimating essential parameter values. Model outputs nicely agree with field measurements during the extreme event, which causes abnormally higher tidal amplitude, etc., demonstrating significant salt intrusion which requires attention from the management side. The manuscript has rich contents but need some work to refine.

Major Comments: ————————

1. Given that the manuscript focus on the effect of "Super-blood-moon", it is necessary to give a clear statement (perhaps a few sentence) to briefly explain the factors from the "super-moon" as well as the "blood-moon". These are separate events, which coincidently happened together on Sep. 28 2015. The effect on tides from the "blood-moon" comes from the alignment of Sun-Earth-Moon, indicated by the total eclipse. But the effect on tides from the "super-moon" comes from the minimum distance between Earth and Moon, which gives extra gravitational pulling. In this sense, the first paragraph of the introduction should be reorganized to reflect both points, and later in sect. 7 "Total lunar eclipse impact", the first sentence needs to be revised as the eclipse is mismatched with the statement of "a moon closest to the earth". Actually, since lunar eclipse only occurs right around full moon, and supermoon traditionally refers only to new moon or full moon at closest point, the "blood-moon" itself is repetitive concept here - but I guess it is just a minor issue and can be ignored.

2. In Sect.2, the authors provided a detailed, yet lengthy review on the analytical and numerical models. The efforts putting in listing and organizing the derivation of equations are definitely appreciated - it is understandable that keeping track of all the variables must be a difficult task. However, I'm afraid that it is necessary for the authors to provide a nomenclature to make it easier for the readers as well as to avoid errors of giving the same physical parameter two different names (such as Qf and Q, and the one of manning's coefficient pointed out by the other referee). Another request, is to have a definition for each symbol when it first appears in the text, such as "a" in equation (4). Between sect.2 and sect.5, it would be nice to directly refer to the equations in sect.2 by number when discussing the results in sect.5. This is also to make sure that the listing of equations in sect.2 is not too excessive - in fact I think some equations can be removed from the text if they are not directly connected to the

modeling work later.

3. Perhaps I have not fully understood the meaning of "predictive" power of the analytical models - as it seems to me the calibration (and optimization of parameters) is applied to all the data, including the normal tides and the supermoon eclipse. The Levenberg-Marquardt method applied here essentially fit the curves to the measured salinity data. Then the question is: how is this able to provide prediction if it needs to be calibrated for each event (change of tidal amplitude, etc.) using measured data? On the other hand, the authors did a nice job explaining the power of the numerical model, by calibrating the model using only one day of measurements within the 3-day data, and validated the model with the other two days' data. Please at least comment on how the analytical models can apply to other extreme events for which no direct measurements are available. And, a brief comment on the difference between the curves produced from computed parameter values (Eq.8,9) and produced with optimized values would be helpful to further demonstrate the power of analytical models.

4. Following the points above, can the authors expand a little bit on the application to other extreme events? Seems that the effect on salinity change comes from abnormally high tides caused by the celestial event, how would these model be applied to those events? Most importantly, could the authors comment on how and where to obtain the necessary information/data to calibrate the model?

5. This manuscript contains a lot of good work, but the grammar and format need a thorough improvement. A few key things include: - The section/sub-section numbers are wrong, e.g. section 2 starts with subsection 1.2, which should be 2.1 I assume. - The "product" symbol (the dot) in the equations are in the wrong format. Also please be careful to unify the format of all mathematical symbols in the text, e.g. all in italic, etc. - There are too many parentheses in the text - some should be removed and some would read better if the text in the parentheses is re-written into the main text. I point out a few places in the technical corrections below. - Please check up the text carefully in the revision to minimize mis-spelling, etc.

Minor Comments: ———— Page 1, Line 18 The parenthesis around Savenije theory should be removed.

Page 3, Line 26 "downstream" should be replaced by "seaward".

Page 3, Line 27 "where" should be removed.

Page 4, Line 19 The parenthesis around "equation (2.38)" should be removed.

Page 8, Line 17 "hydrodynamic radius" should be "hydraulic radius". Hydrodynamic radius is a completely different concept that describe particles moving through solute.

Page 8, Line 22 Remove ", n is the"

Page 10, Line 21 please provide the source/reference of the bathymetry data.

Page 11, Line 6 "the depth gradually increases", in which direction. Also the same paragraph is confusing to read.

Page 11, Line 9 A comment on how these stations are chosen would be great. Was it for the availability or it has taken into account the best location for model calibration/validation?

Page 11, Line 21 Remove parentheses around "spring tide"

Page 12, Line 11 Remove parentheses around "07th-...2015".

Page 12, Line 29 As pointed out by the other referee, "dumping" should be "damping".

Page 13, Line 15 Please explain the meaning of "predictive uncertainty of these equations" (refer to main comment #3).

Page 14, Line 13-15 The source of data for plotting Fig.13 is unclear - please provide a brief explanation.

Page 15, Line 1 "valided" should be "validated"

Page 15, Line 13-21 These two paragraphs should be merged and remove repetitive

statements. Instead of saying "of a selected system parameter", directly stating "of the dispersion coefficient" and removing the sentence in Line 15 will make the text more streamlined and easier to follow.

Page 17, Line 15 The statement is inaccurate. The impact of eclipse should be coming from the lineup of earth, sun and moon, that Earth is exactly in the middle of the moon and the sun. "supermoon" refers to a moon closest to the Earth, which is a separate event. Although in 2015 the concurrence of the two phenomenon is very rare.

Page 17, Line 16-18 Please give the specifics (describe in words, perhaps a few sentences listing key points) of the "significant impact".

Figure 6 It would be nice to have a scale for the inset figure on the right side.

Figure 9 The symbols in the figures in the center row are very difficult to tell apart.

---

## Author Comment (AC1) · 27 Aug 2016

**Dear Reviewer #1,**

Thank you very much for your helpful review. We have carefully studied the comments and suggestions and revised our paper accordingly. The following are our point-by-point responses to the general and specific comments. We hope that our responses adequately address the comments. Below is our response to the issues raised in the review.

*This is a well-prepared and well-documented paper. Besides some minor editing that is required, the paper is well written. I presume the publisher will take care of the English editing and minor corrections. (just as a reminder, please make sure to replace dumping by damping in P.12, L29.*

**Our reply:** Some minor corrections have been realized and the English style improved in the revised MS in response to the reviewer's comments.

*The case of analysing an extreme spring tide in an intensively used estuary is very relevant. It is definitely a cutting-edge case study. The study has been very well done, and it is based on a very detailed data set of intensive measurements. This makes this paper very interesting and relevant for HESS.*

**Our reply:** Thank you for your thorough review and salient observations. It is our sincere hope that this paper provides the necessary science.

*In Figure 10, and also in Table 5, we three moments in time for HWS, TA and LWS. This is strange, because at every cycle, there is only one moment of HWS, LWS and TA. Is this because the authors did not observe the moment of slack, but just derive it from the temporal observation of the salinity (as in Figure 6). But if that is so, than the maximum value corresponds to HWS, the minimum value to LWS and the time-average value to TA. There should not be multiple values. Also in Figure 10, I don't understand why the differences between the observations that are only 20 min apart are so large. In Figure 6 the maximum values of the salinity curves are rather flat. So the differences should not be large.*
*Maybe the authors determined the moment of HWS on the basis of the hydraulic model. But that would be wrong, since the hydraulic model may determine the moment of slack wrongly. The correct moments of slack, if not observed in the field should correspond with the maximum and minimum observed salinities.*

**Our reply:** We agree with the comments from reviewer that there is only one moment of slack for HWS and LWS at every cycle, which occurs when the water velocity is lower, starting so many minutes after HW and LW. In our case, the measured velocity for slack time is very minimal (less than 0.05m/s). Experimentally, water velocity was measured using cords attached to floats. In addition, the slack duration was approximately 40 min at HWS and 47 min at LWS. The moment of TA occurrence was estimated in reference to HWS and LWS occurrence. The above curve in Fig.10 at HWS and the below curve at LWS during 1$^{st}$ and 2$^{nd}$ cycle correspond to the two slack

moments. The two other curves (at HWS and LWS) have been incorporated in the old version of the MS only to help us identifying the exact slake moments. In the revised paper we shows just the curves for the exact HWS and LWS slack moments (see figure 10 below), which corresponds respectively to the maximum and the minimum of the observed salinities (figure 6).

**Minor comments:**

If the hydraulic model is calibrated on the Roughness, then it is useless to present the composite equation for the Manning roughness (16).
Similarly, if the dynamic 1-D salt balance equation is calibrated on the Dispersion, then don't mention (19) in the paper. Moreover equation (19) is not at all appropriate for salt intrusion. It refers to rivers only. So eqs (16) and (19) should be removed from the paper.
Also use the same parameters throughout the paper. So if you use K_manning in one equation then don't use n in another.

**Our reply:** The Manning-Strickler friction coefficient ($K_s$) is clarified in revised MS (i.e., **Ks** is the inverse value of n (**$K_s=1/n$**), n: Manning's coefficient). Authors think the Manning-Strickler equation can help the readers to well understand the calibration process. Authors have no problem to remove this equation if requested in the final stage of the manuscript editing. The equation (19) is removed in the revised MS.

[Figure]

Fig. 10- Observed and analytically computed longitudinal salinity distribution along the Sebou estuary (surveyed on 28$^{th}$ September 2015) during 1$^{st}$ cycle (a, b, c) at LWS, TA and HWS, and 2$^{nd}$ cycle (d,e,f) at LWS TA and HWS.

---

## Author Comment (AC2) · 27 Aug 2016

Reponses to comments by Reviewer #2

**Reponses to comments by Reviewer #2**

(Reviewer's comments are shown in *Italic*)

**Dear Reviewer #2;**

Thank you very much for your helpful review. We have carefully studied the comments and suggestions and revised our paper accordingly. The following are our point-by-point responses to the general and specific comments. We hope that our responses adequately address the comments. Below is our response to the issues raised in the review.

*1. Given that the manuscript focus on the effect of "Super-blood-moon", it is necessary to give a clear statement (perhaps a few sentence) to briefly explain the factors from the "super-moon" as well as the "blood-moon". These are separate events, which coincidently happened together on Sep.28 2015. The effect on tides from the "blood-moon" comes from the alignment of Sun-Earth-Moon, indicated by the total eclipse. But the effect on tides from the "super-moon" comes from the minimum distance between Earth and Moon, which gives extra gravitational pulling. In this sense, the first paragraph of the introduction should be reorganized to reflect both points, and later in sect. 7 "Total lunar eclipse impact", the first sentence needs to be revised as the eclipse is mismatched with the statement of "a moon closest to the earth". Actually, since lunar eclipse only occurs right around full moon, and super-moon traditionally refers only to new moon or full moon at closest point, the "blood-moon" itself is repetitive concept here - but I guess it is just a minor issue and can be ignored.*

**Our reply (1):** Thank you very much for all your constructive comments, which have been used to improve the quality of our manuscript. As regards your general observation, we have added the following paragraph in the Introduction Section: *During this event, three things will occur at once. The moon will be both full and at its closest point to Earth (356,877 Km) away; that's known as a Super-Moon or Perigee Moon (NASA, 2015). And this will occur at the same time as a total lunar eclipse; that means the Moon, Sun and Earth will be aligned. Because of its proximity to Earth, the Moon will appear brighter and larger (14% larger and 30% brighter than other full moons) in the sky (NASA, 2015). And it will appear a dark, coppery red, caused by the Earth blocking the sun light that refracted by the atmosphere into the umbra (Hughes et al., 2015). In the other hand, the tidal motions are controlled by changes in the position and alignment of the Moon and Sun relative to Earth (Stronach, 1989). In addition, tidal forces are strengthened if the moon is closest to Earth in its elliptical orbit and when the Sun and Moon are directly over the equator (NOC, 2015).*

**Our reply (2):** Similarly, we have included the following sentence in the Total lunar eclipse section: *The impact of the combination of a Super-Moon and a total lunar eclipse on river hydrodynamics are mainly caused by the moon closest to Earth, which gives extra gravitational pulling, and by the alignment of Sun-Earth-Moon.*

**Our reply (3):** We agree with your good observation that the "Super-Moon" traditionally refers only to new moon or full moon at closest point, the "blood-moon" is a repetitive appellation. We have revised this term in all text.

*2. In Sect.2, the authors provided a detailed, yet lengthy review on the analytical and numerical models. The efforts putting in listing and organizing the derivation of equations are definitely appreciated - it is understandable that keeping track of all the variables must be a difficult task. However, I'm afraid that it is necessary for the authors to provide a nomenclature to make it easier for the readers as well as to avoid errors of giving the same physical parameter two different names (such as $Q_f$ and $Q$, and the one of manning's coefficient pointed out by the other referee). Another request is to have a definition for each symbol when it first appears in the text, such as "a" in equation (4). Between sect.2 and sect.5, it would be nice to directly refer to the equations in sect.2 by number when discussing the results in sect.5. This is also to make sure that the listing of equations in sect.2 is not too excessive - in fact I think some equations can be removed from the text if they are not directly connected to the modeling work later.*

**Our reply:** We thank the reviewer for the detailed reading. In the revised MS; we have included a Nomenclature and an Abbreviation lists. In addition, parameters nomenclature was clarified and homogenized along the paper: for example the Manning-Strickler friction coefficient ($\mathbf{K_s}$) is clarified as the inverse value of Manning's coefficient n ($\mathbf{K_s=1/n}$).

In the other hand, in our view the presentation of the equations in Sec.2 (for the analytical or numerical model) is essential for a best understanding of the theoretical content. Authors have no problem in changing this section if requested in the final stage of the manuscript editing.

*3.Perhaps I have not fully understood the meaning of "predictive" power of the analytical models - as it seems to me the calibration (and optimization of parameters) is applied to all the data, including the normal tides and the supermoon eclipse. The Levenberg-Marquardt method applied here essentially fit the curves to the measured salinity data. Then the question is: how is this able to provide prediction if it needs to be calibrated for each event (change of tidal amplitude, etc.) using measured data?*

*On the other hand, the authors did a nice job explaining the power of the numerical model, by calibrating the model using only one day of measurements within the 3-day data, and validated the model with the other two days' data. Please at least comment on how the analytical models can apply to other extreme events for which no direct measurements are available.*

*And,a brief comment on the difference between the curves produced from computed parameter values (Eq.8, 9) and produced with optimized values would be helpful to further demonstrate the power of analytical models.*

*4. Following the points above, can the authors expand a little bit on the application to other extreme events? Seems that the effect on salinity change comes from abnormally high tides caused by the celestial event, how would these models be applied to those events? Most importantly, could the authors comment on how and where to obtain the necessary information/data to calibrate the model?*

**Our reply (1):** Thanks for raising this important point. "Predictive models" are very interesting tools for researchers, engineers and water managers to obtain first order estimates of essential process in estuaries. In addition, the analytical predictive models is completely transparent, practical and *requiring minimal data*, allowing direct assessment of the influence of individual variables and parameters on the salinity intrusion. The analytical model of salt intrusion used in this paper merits well the appellation of *Predictive tool* for many reasons:

**1-** Using limited data of salinity in the seaward part of the river at HWS permits the calculation of the parameters $S_0$, $D_0^{HWS}$ and K. Consequently, the salinities and dispersion coefficients at any point along the estuary can be easily *predicted* using equations 1 and 2.

$$\frac{D^{HWS}}{D_0} = \left[\frac{S}{S_0}\right]^K \tag{1}$$

$$\frac{D^{HWS}}{D_0} = \left[1 - \frac{K|Q|a}{AD_0}\left\{\exp(x/a) - 1\right\}\right] \tag{2}$$

**2-** If the model is calibrate using data measurements achieved at HWS, the salinity distribution can be computed at LWS and TA based on the relation between salinity distributions during the three conditions. The salt distribution curve at HWS could be shifted downstream over a horizontal distance equal to the tidal excursion length (E) and half of the tidal excursion length (E/2) to *predict* the salt distribution curve at LWS and TA conditions respectively (Savenije, 2012).

[Figure]

**3-** The eclipse during September 28$^{th}$ 2015 caused an increase in water level and salinity concentration along the Sebou estuary. The study of saline intrusion at this event permit

the realization of salinity nomograms (salinity profile for different river flows), based on the K and D parameters calibrated for this event. If extreme and similar situations occur in the future (other eclipse, global sea rise), managers can use these nomograms to get a quick estimation of salt intrusion length and the salt amplitude, even with limited measurements. The K and D settings for these predicted events will be different but very closest to those of the 2015's eclipse (same family and event category). The salinity in the estuary during future extreme events can be calculated with a good error range.

For example, according to the 5th report of Intergovernmental Panel on Climate change (IPCC, 2014) Moroccan costal Atlantic sea level will increase significantly during the next decades. Sea level rise can severely affected Fresh-water resources and push saline water further upstream in estuaries. Sea level rise that will be a consequence of global warming is similar to that observed during the short extreme event of 2015's eclipse. Studying the eclipse of 2015 can help to predict future sea level rise impacts.

*And, a brief comment on the difference between the curves produced from computed parameter values (Eq.8, 9) and produced with optimized values would be helpful to further demonstrate the power of analytical models.*

**Our reply (2):** *The curves produced from computed parameters, i.e.,Van der Burgh's coefficient K and dispersion coefficient $D_1$ presents a good correlation with the curves from optimized values (see figure below). This indicates that the predictive equations developed by Savenije (1993a, 2012) and revised by Gisen et al (2015) are appropriate to be applied in getting a first estimate of K and $D_1$ and starting values for the optimization procedure (quick convergence).* Authors have no problem in adding these figures if requested in the final stage of the manuscript editing.

[Figure]

Figure. Observed and analytically model for longitudinal salinity distribution along the Sebou estuary at HWS

*5. This manuscript contains a lot of good work, but the grammar and format need a thorough improvement. A few key things include: - The section/sub-section numbers are wrong, e.g. section 2 starts with subsection 1.2, which should be 2.1 I assume. - The "product" symbol (the dot) in the equations are in the wrong format. Also please be careful to unify the format of all mathematical symbols in the text, e.g. all in italic, etc. - There are too many parentheses in the text - some should be removed and some would read better if the text in the parentheses is re-written into the main text. I point out a few places in the technical corrections below. - Please check up the text carefully in the revision to minimize mis-spelling, etc.*

**Our reply:** Thank you for pointing this out. In the revised MS, we have organized all Section/Sub-section and the "product" symbol "dot" in the equations is removed. Also the many parentheses in the text some are removed. In addition, all mis-spelling and grammatical errors pointed out by the reviewers have been corrected.

**Minor Comments:**

*Page 1, Line 18 The parenthesis around Savenije theory should be removed.*

**Our reply:** Corrected, made in the revised MS.

*Page 3, Line 26 "downstream" should be replaced by "seaward".*

**Our reply:** Corrected, made in the revised MS.

*Page 3, Line 27 "where" should be removed.*

**Our reply:** Corrected, made in the revised MS.

*Page 4, Line 19 The parenthesis around "equation (2.38)" should be removed.*

**Our reply:** Corrected, made in the revised MS.

*Page 8, Line 17 "hydrodynamic radius" should be "hydraulic radius". Hydrodynamic radius is a completely different concept that describe particles moving through solute.*

**Our reply:** We agree with your comments. Corrected, made in the revised MS.

*Page 8, Line 22 Remove ", n is the".*

**Our reply:** Corrected, made in the revised MS.

*Page 10, Line 21 please provide the source/reference of the bathymetry data.*

**Our reply:** The bathymetric data used in the study (analytical and numerical models) were provided by local water authorities (i.e. ANP, National Agency of Ports and Water Services of Kenitra town) in a row format (AutoCAD files). In other hand, some topographic and satellite maps were exploited. This information have been indicated in the revised MS.

*Page 11, Line 6 "the depth gradually increases", in which direction. Also the same paragraph is confusing to read.*

**Our reply:** The depth gradually increases from Lalla Aïcha dam (upstream) to seaward. Also, we have rephrased this paragraph in the revised MS.

*Page 11, Line 9 A comment on how these stations are chosen would be great. Was it for the availability or it has taken into account the best location for model calibration/ validation?*

**Our reply:** These stations were chosen according to:
1- Salinity classification (zones) according to Venice System (Symposium on the classification of brackish waters, Venice, 1958; Haddout et., 2016).
2- The availability of data (hydraulic and salinity) in some stations. These stations were used in models calibration/validation.
3- The accessibility.

*Page 11, Line 21 Remove parentheses around "spring tide".*

**Our reply:** Corrected, made in the revised MS.

*Page 12, Line 11 Remove parentheses around "07th-...2015".*

**Our reply:** Corrected, made in the revised MS.

*Page 12, Line 29 As pointed out by the other referee, "dumping" should be "damping".*

**Our reply:** Yes, corrected, made in the revised MS.

*Page 13, Line 15 Please explain the meaning of "predictive uncertainty of these equations" (refer to main comment #3).*

**Our reply:** It is just a repetition error. Corrected in the MS.

*Page 14, Line 13-15 The source of data for plotting Fig.13 is unclear - please provide a brief explanation.*

**Our reply:** The source of data for plotting Fig.13 is: Savenije (2005) *Salinity and Tides in Alluvial Estuaries,* Chapter 5: Salt intrusion in alluvial estuaries, p.171-173. This information have been added to the Fig.13 title.
We have added the following sentences in the revised MS: Figure 13 shows *the salt intrusion lengths computed against observed lengths (at HWS) with data specific to Sebou estuary compared to different predictive formulae found in the literature. We observe that Savenije solution for salt intrusion lengths fits very well the observed data compared to all others solutions. The results of Savenije model are considered most accurate.*

*Page 15, Line 1 "valided" should be "validated".*

**Our reply:** Corrected, and made in the revised MS.

*Page 15, Line 13-21 These two paragraphs should be merged and remove repetitive statements. Instead of saying "of a selected system parameter", directly stating "of the dispersion coefficient" and removing the sentence in Line 15 will make the text more streamlined and easier to follow.*

**Our reply:** Thank you for pointing this out. We have rephrased this sentence in the revised MS.

*Page 17, Line 15 The statement is inaccurate. The impact of eclipse should be coming from the lineup of earth, sun and moon, that Earth is exactly in the middle of the moon and the sun. "supermoon" refers to a moon closest to the Earth, which is a separate event. Although in 2015 the concurrence of the two phenomenon is very rare.*

**Our reply:** We agree with your comment, we have added the sentences: "*The impact of the combination of a Super-Moon and a total lunar eclipse on river hydrodynamics are mainly caused by the moon is closest to Earth, which gives extra gravitational pulling, and the alignment of Sun-Earth-Moon*".

*Page 17, Line 16-18 Please give the specifics (describe in words, perhaps a few sentences listing key points) of the "significant impact".*

**Our reply:** *The maximum salinity at high water along the Sebou estuary has been described in section 4. Super-Moon and total lunar eclipse impact on the maximum salinity at different locations compared to the normal situation is given in Table 10. The results clearly show that the astronomical event impact on salinity intrusion is highly significant. The salinity increments in the four stations relative to the normal situation were respectively 4.1g/l (6.54%), 9.4g/l (22.06%), 0.5g/l (26.32%) and 0.2g/l (14.29%). This situation was unsuitable for drinking and agriculture propose.*

*Figure 6 It would be nice to have a scale for the inset figure on the right side.*

**Our reply:** Yes, in revised MS we added a scale for the inset figure.

*Figure 9 The symbols in the figures in the center row are very difficult to tell apart.*

**Our reply:** The figure 9 is changed in the revised MS according to this comment.

**Reference**

Stronach, J. A., Murty, T. S.: Nonlinear river-tidal interactions in the Fraser River, Canada.Marine Geodesy, 13(4), 313-339. 1989.

Venice System. Symposium on the classification of brackish waters, Venice, April 8–14, 1958, Archives for Oceanography and Limnology 11 (Suppl.), 1958, pp. 1-248

Haddout, S., Maslouhi, A., Magrane, B., & Igouzal, M. (2016). Study of salinity variation in the Sebou River Estuary (Morocco). Desalination and Water Treatment, 57(36), 17075-17086.

IPCC 2014: Intergovernmental Panel on Climate Change. Climate Change 2014–Impacts, Adaptation and Vulnerability: Regional Aspects. Cambridge University Press. (2014).

National Aeronautics and Space Administration (NASA), 2015. http://www.nasa.gov/ (last access: 18.08.16).

National Oceanography Centre (NOC), 2015. http://noc.ac.uk/ (last access: 18.08.16).

Savenije, H.H.G.: Salinity and Tides in Alluvial Estuaries, second ed. <www. salinityandtides.com> (last access: 08.10.15). 2012.

Savenije, H.H.G.: Salinity and tides in alluvial estuaries. Amsterdam: Elsevier, 197 pp. 2005.

Savenije, H.H.G.: Predictive model for salt intrusion in estuaries. Journal of Hydrology, 148(1), 203-218. 1993a.

Gisen, J. I. A., Savenije, H. H. G., and Nijzink, R. C.: Revised predictive equations for salt intrusion modelling in estuaries. Hydrology and Earth System Sciences, 19(6), 2791-2803. doi:10.5194/hess-19-2791-2015. 2015b.